# On the relative motions of long-lived Pacific mantle plumes

Kevin Konrad[1], Anthony A.P. Koppers[1], Bernhard Steinberger [2,3], Valerie A. Finlayson[4], Jasper G. Konter[4] & Matthew G. Jackson[5]

Mantle plumes upwelling beneath moving tectonic plates generate age-progressive chains of volcanos (hotspot chains) used to reconstruct plate motion. However, these hotspots appear to move relative to each other, implying that plumes are not laterally fixed. The lack of age constraints on long-lived, coeval hotspot chains hinders attempts to reconstruct plate motion and quantify relative plume motions. Here we provide $^{40}Ar/^{39}Ar$ ages for a newly identified long-lived mantle plume, which formed the Rurutu hotspot chain. By comparing the inter-hotspot distances between three Pacific hotspots, we show that Hawaii is unique in its strong, rapid southward motion from 60 to 50 Myrs ago, consistent with paleomagnetic observations. Conversely, the Rurutu and Louisville chains show little motion. Current geodynamic plume motion models can reproduce the first-order motions for these plumes, but only when each plume is rooted in the lowermost mantle.

[1] College of Earth, Ocean, and Atmospheric Sciences, Oregon State University, Corvallis, OR 97331, USA. [2] GFZ German Research Centre for Geosciences, 14473 Potsdam Germany. [3] Centre for Earth Evolution and Dynamics (CEED), University of Oslo, 0315 Oslo, Norway. [4] Department of Geology and Geophysics, School of Ocean and Earth Science and Technology, University of Hawaii Manoa, Honolulu, HI 96822, USA. [5] Department of Earth Science, UC Santa Barbara, Santa Barbara, CA 93106, USA. Correspondence and requests for materials should be addressed to K.K. (email: Konradke@oregonstate.edu)

Ever since the hypothesis that mid-plate ocean island chains represent the surface expression of deep seated mantle plumes[1,2], their linear age progressions have been used to infer past plate motion[3,4]. However, efforts to constrain Pacific plate motion are complicated by the potential effects of independent plume motion[5–8]. Paleomagnetic paleolatitude data from four seamounts in the Emperor chain were interpreted as showing a large southward Hawaiian plume drift of ~15° relative to Earth's spin axis, from 80 to 50 Myrs ago[8]. In contrast, similar reconstructions of the Louisville hotspot indicate little (<4°−5°) to no latitudinal movement during the same time frame and thus eliminate the idea of a large-scale constant southward mantle flow[6]. In fact, recent geodynamic modeling suggests that the Hawaiian plume may be unique in its behavior. Focused subduction along the Aleutian arc between 100 and 50 Ma may have caused a strong southward deep mantle flow that has affected the Pacific large low shear velocity province (LLSVP)[9] and the starting locations of plumes, assuming that plumes initiate from the LLSVP edges[10]. Alternatively, hypotheses for Hawaiian plume motion may invoke capture and subsequent release of the

Hawaiian plume by the ancient Kula-Pacific ridge during the 100–50 Ma time interval[7,8], or asthenospheric flow channeling[11,12] towards a spreading ridge north of Hawaii[7,8]. In an effort to better constrain the degree to which hotspots move independently of each other in the Pacific mantle domain, we compare the age and geometry of the Hawaiian and Louisville chains to the newly mapped, intermittently expressed, Rurutu chain in the Mid-Pacific (Fig. 1). This allows for direct observations in the variation of plume motion among three contemporaneous hotspots that were active beneath the Pacific plate, from approximately 72 Ma to the present day.

The Rurutu hotspot[13,14] is currently active beneath the young Arago Seamount (~230 ka[15]) and contains a distinct isotopic character that varies from HIMU (high $\mu = {}^{238}U/{}^{204}Pb$ in source component[16]) to C (common mantle component[17]) that was originally identified as the "Atiu Trend"[18] in the Cook-Austral islands. Here we provide new age constraints on the Tuvalu seamount chain in the mid-west Pacific between 3° S, 175° E and 10° S, 180° (Fig. 1, Supplementary Fig. 1). The seamounts and islands in this region are oriented roughly northwest-southeast,

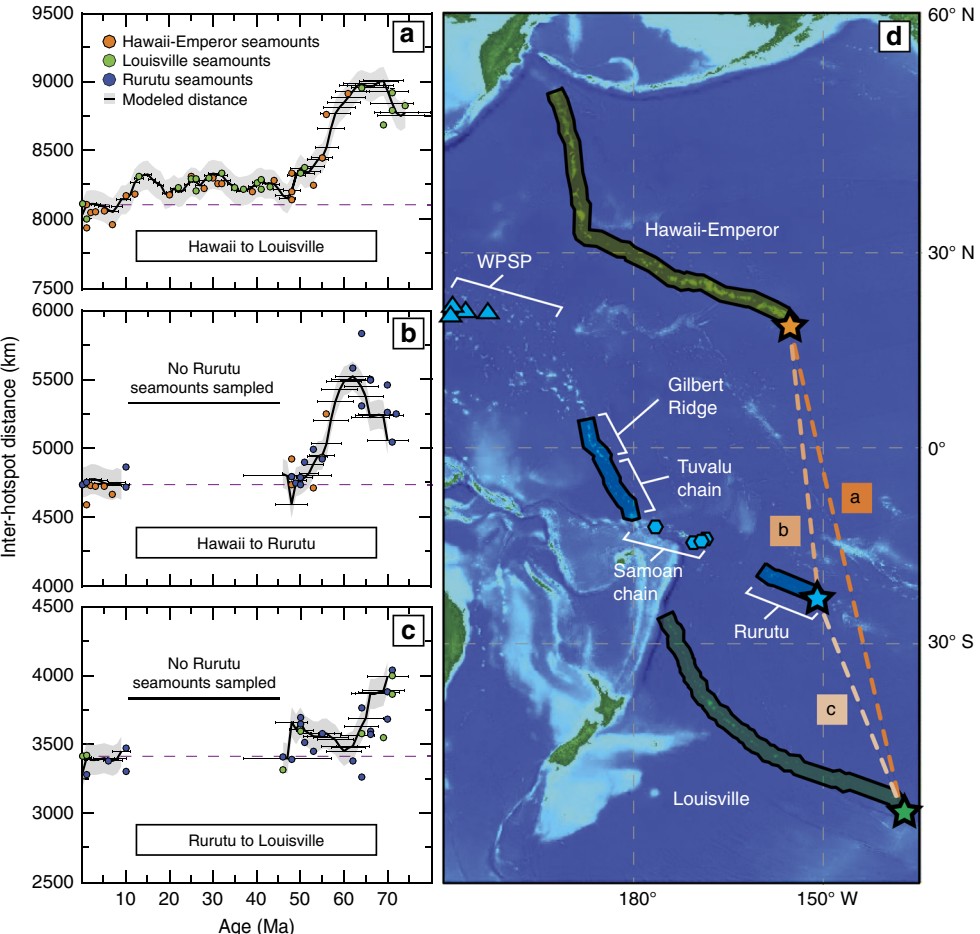

**Fig. 1** Inter-hotspot distances as a function of age and the geographic locations of the Hawaii and Louisville and Rurutu hotspots. **a–c** The black lines represent the distance between the two compared model hotspot chains (see Methods) and a dashed purple line displays the modern day inter-hotspot distance. One sigma uncertainties are provided for reconstructed model ages at 1 Ma increments and the gray shading represents distance uncertainties assuming a plume radius of 75 km. The circles represent the distance between a seamount of a given age and the point where a coeval modeled seamount falls geographically on the compared hotspot chain, confirming the estimated uncertainty bounds on the inter-hotspot distances. The inferred center of a seamount was used for the geographic location. **a** Hawaii compared to Louisville; **b** Hawaii compared to Rurutu; **c** Rurutu compared to Louisville. **d** A digital elevation map (ETOPO1)[71] of the western Pacific showing the modeled reconstructions of the Hawaii-Emperor (orange), Rurutu (blue), and Louisville (green) chains. Stars denote the current hotspot locations presumed at Loihi Seamount[72], Arago Seamount[15], and the inferred Louisville hotspot location[35]. Blue hexagons represent the location of seamounts with Rurutu-like geochemical compositions[34] but lack age determinations. Blue triangles represent HIMU seamounts within the Western Pacific Seamount Province (WPSP) that contain ages consistent with belonging to the Rurutu chain[13,32]

located between the Gilbert Ridge and Samoan hotspot chain, and rest upon lithosphere generated during the Cretaceous paleo-magnetic super normal (~110–85 Ma)[19]. The subaerial portions of the islands are entirely composed of coral and thus there have been no previous geochemical and geochronological data on basalts reported from these islands.

The new $^{40}$Ar/$^{39}$Ar age determinations presented here link the Tuvalu seamount chain to the Rurutu hotspot, allowing us to define the motion of the Rurutu plume relative to the previously dated, long-lived Louisville and Hawaiian hotspots. We then compare these relative motions with the outcome from numerical plume motion models[20–23] in order to test which geodynamic mantle parameters best reproduce the observed motions. Results indicate that the Hawaiian plume is unique in its rapid motion ~60–50 Myrs ago, while the Rurutu and Louisville plumes appear to be either relatively stationary or moving in tandem. We find that modeled motions can reproduce first-order trends only if plumes are rooted, but not anchored, near the core-mantle boundary (CMB).

## Results

**Longevity of the Rurutu hotspot chain**. Seamounts from the Tuvulu region, mid-Pacific (Fig. 1) were dredged during the RR1310 expedition. In most dredges, basaltic material was recovered that has been variably altered from which a selective number of samples were deemed useable for $^{40}$Ar/$^{39}$Ar age determinations (see Supplementary Note 1). These seamounts range in age from 64 to 47 Ma (Supplementary Table 1, Supplementary Note 2) and have a distinctive HIMU isotopic character with $^{206}$Pb/$^{204}$Pb values greater than 20 (Supplementary Table 2). The ages and chemistry of the Tuvalu seamounts fit ideally within the age progression of the Rurutu hotspot chain (see Supplementary Discussion, Supplementary Figs. 2–4), for which a short 10−0 Ma segment is documented in the Cook-Austral islands region[15,24–30]. Another 72–64 Ma segment of this hotspot chain is located in the Gilbert Ridge[14,31] and HIMU seamounts can be traced even farther into the Western Pacific Seamount Province (WPSP) back to 120 Ma[13,14,32,33]. However, a lack of sampling of the seamounts between the Cook-Austral islands and Samoa, and the lack of datable material from four seamounts that carry a Rurutu composition within the Samoan archipelago[34], limit our mapping of this hotspot chain between 48 and 10 Ma. It also is important to note that a rapid increase in apparent plate motion observed along the Gilbert Ridge (66–72 Ma)[14,31] (Fig. 1b, c) is potentially a function of preferentially sampling late stage and/or rejuvenated volcanism, because no other seamount chain[35,36] displays such a small age change over more than 700 km of any along-seamount-chain distance. Therefore, it remains uncertain whether the sharp apparent motion of the Rurutu plume between 66 and 72 Ma is a function of independent plume motion or sampling bias. These new age constraints confirm that the Rurutu plume is long-lived[13] and intermittently expressed where litho-spheric weaknesses allow melt penetration.

**Relative inter-hotspot motions**. With the age and geometry of the three longest-lived hotspots mapped out on the Pacific plate (Supplementary Fig. 5, Supplementary Table 3), changes in inter-hotspot distance over time between their mantle plumes can be computed (Fig. 1) that reveal fundamental differences in relative hotspot motions. First, we generated a best-fit age progression along each individual hotspot chain using $^{40}$Ar/$^{39}$Ar age determinations and seamount locations (see Methods). Then we cal-culated great circle distances and uncertainties between combinations of two best-fit chains at any given age (Fig. 1; black lines, error bars). Moreover, by calculating distances between a

seamount of a measured age and its coeval point on the modeled age progression of one of the other two hotspot chains, we pro-duced another proxy for the scatter in these data sets (Fig. 1; circles). The comparison of the Hawaii to Louisville hotspots (Fig. 1a) confirms previous observations[8,36] and shows a large plume divergence (640 ± 106 km) from 60 to 48 Ma, relatively constant inter-hotspot distances between 48 and 15 Ma, and a small decrease in distance starting around 15 Ma. The compar-ison of Rurutu and Hawaii (Fig. 1b) shows a similar decrease in hotspot distance (684 ± 106 km) between 60 and 48 Ma, which is consistent with a fast-moving Hawaiian plume. These inter-hotspot distances correspond to a relative rate of motion for the Hawaiian plume of 53 ± 21 km/Myr (1$\sigma$) as compared to Louis-ville and 57 ± 27 km/Myr as compared to Rurutu during that 12 Myr interval. These rates indicate a faster rate of Hawaiian hotspot motion than previous models have suggested[20,37], if Rurutu and Louisville were approximately stationary in this time interval. Our model-independent method, based solely on chain geometry and radiometric ages, therefore shows that significant motion of the Hawaiian plume occurred, supporting the under-standing that observed changes in paleomagnetic-derived paleo-latitude[7,8] cannot be solely a result of true polar wander[37]. The smaller change in distance between Rurutu and Louisville within the same 12 Myr interval (~200 km; Fig. 1c) implies significantly less inter-hotspot motion. Importantly, this indicates that the Rurutu and Louisville plumes either have been relatively immo-bile or have moved similarly, likely in a mainly eastward drift, as models suggest for the past 70 Myrs[9,20,22]. As paleomagnetic data for Louisville[6] indicate hardly any southern drift, a similar limited latitudinal change in the Rurutu hotspot location is expected. Given that comparable eastward drifts of both the Louisville and Rurutu hotspot are supported by plume motion models in the 66−50 Ma timeframe (discussed further below) we favor this latter interpretation.

**Comparisons to geodynamic plume motion models**. In an effort to test if the different relative motion histories between these three long-lived Pacific plumes can be reproduced, a previously developed geodynamic model[20–23] was employed, while varying key parameters (see Methods for details). In these models, there are three parameters that have major time-dependent controls on large-scale mantle flow and the lateral advection of mantle plumes. These major controls include the mantle density gradients that can be inferred from seismic tomographic models, the assumed mantle viscosity structure, and the assigned relative viscosity of the plume conduits that controls plume upwelling velocities. Finally, the starting age, buoyancy flux, depth of the plume root, and whether or not that root is mobile or anchored, also will affect the predicted indi-vidual plume motions. We varied all these parameters, including a suite of tomography models and published viscosity structures, in our model runs to best reproduce the observed relative motions of the Hawaii, Louisville, and Rurutu hotspots. In our approach, we applied the plate motion model of Torsvik et al.[38] that for Pacific plate motions are based on rotation poles of Steinberger and Gaina[39]. Even though the Steinberger geo-dynamic models[20–23] are limited by using tomographic models of the modern mantle to predict past large-scale mantle flow patterns, the insertion of vertically buoyant plumes at repro-ducible locations in every model run allows us to compute tens of thousands of inter-hotspot motion histories. These Stein-berger model outcomes are directly comparable to observed inter-hotspot distances between the three longest-lived hotspot chains in the Pacific (Fig. 2). The Hassan et al.[9] modeling approach, while being more advanced numerically, lacks this

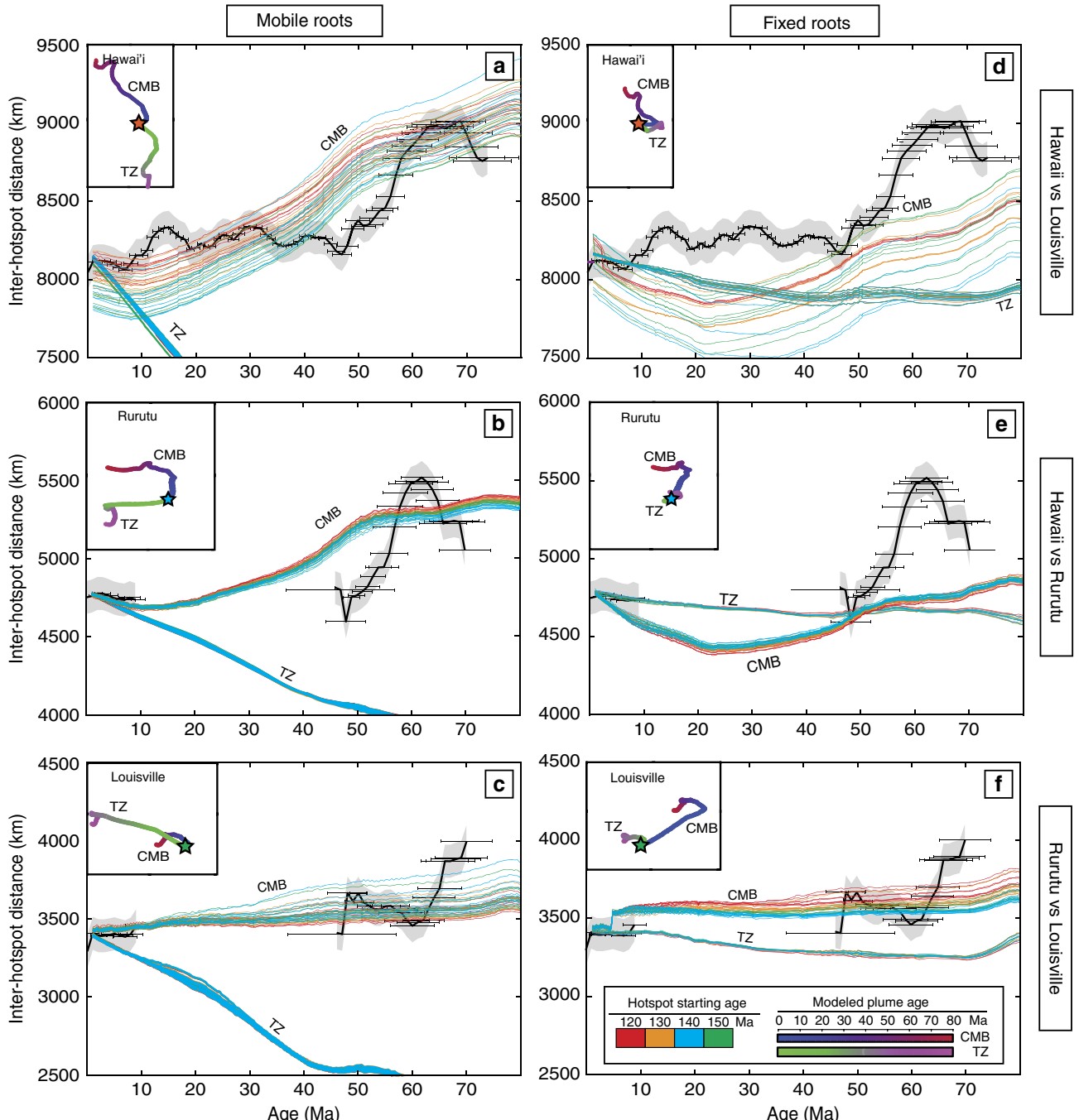

**Fig. 2** The observed relative inter-hotspot distances through time compared against modeled simulations. The relative motions are from Fig. 1 compared with plume motion model results (colored lines) for both plumes rooted in the D″ above the core mantle boundary (CMB) and at a depth of 660 km at the base of the transition zone (TZ). Black lines and gray uncertainty bounds are the measured inter-hotspot distances as shown in Fig. 1. Individual colored lines represent different buoyancy fluxes (1–9×10³ kg/s) and each line color represents a different starting age of the compared hotspot (from 120 to 150 Ma) with the scale shown in the bottom right. The mantle viscosity model used here is from Steinberger and Calderwood[56] and the tomography model used is TOPOS362d1[69]. The starting age for Hawaii is 130 Ma with a buoyancy of 5×10³ kg/s in panels **a**, **b** and **d**, **e** with variable Louisville and Rurutu ages and buoyancy, while it is 120 Ma and a buoyancy of 5×10³ kg/s for Louisville in panels **c** and **f** with Hawaii and Rurutu being varied. **a** Hawaii compared against Louisville with mobile roots. **b** Hawaii compared against Rurutu with mobile roots. **c** Rurutu compared against Louisville with mobile roots. **d** Hawaii compared against Louisville with fixed roots. **e** Hawaii compared against Rurutu with fixed roots. **f** Rurutu compared against Louisville with fixed roots. Example model results for CMB (red-blue) and TZ (pink-green) hotspot locations through time with a starting age of 150 Ma (Hawaii and Rurutu) and 120 Ma (Louisville) and buoyancy fluxes of 5×10³ kg/s (Hawaii and Rurutu) and 3×10³ kg/s (Louisville) are shown as insets in each graph

feature and thus cannot be easily calibrated against our new
$^{40}Ar/^{39}Ar$ geochronological data from the Rurutu hotspot chain
and the observed changes in inter-hotspot distance between
Hawaii, Louisville, and Rurutu over the last 70 Myrs.

The geodynamic plume motion models, to the first order, fit
the magnitude of the inter-hotspot distances, and we make
several key observations. First, in any scenario wherein a plume is
rooted at the boundary between the upper and lower mantle

(~660 km) at the base of the so-called transition zone (TZ), the motions of the plume are significantly reduced or, more commonly, advected laterally in the opposite direction to what is required to match the observed inter-hotspot distances (Fig. 2). On the contrary, our results are in agreement with long-standing geochemical arguments[40,41] and recent seismic evidence[42,43] for a lower mantle origin for these hotspots near the CMB. This infers that hotspots situated within the French Polynesia and Cook-Austral region also may represent primary hotspots as opposed to secondary "plumelets" originating from the shallow top of the mid-Pacific Superplume[32,44]. This is confirmed in model runs that have the Rurutu plume originating in the TZ, while rooting the Hawaiian and Louisville plumes at the CMB, which in the majority of the cases do not match first-order features in the inter-hotspot distance evolutions (Fig. 3). A second observation is that most tomography and viscosity models utilized in this study cannot reproduce the shorter timescale changes in motion between plumes, most specifically the sharp southern drift of the Hawaiian plume between approximately 60–50 Ma. This fast plume motion likely requires some regional mechanism that could be related to plume-ridge capture[7,8] or changes in ancient subducting slab configurations[9] that are not fully resolved in the tomographic models used in our calculations. The third observation is that, when the roots of the plumes are advected with flow in the lower mantle, inter-hotspot motions are generally better represented than when the roots are kept at fixed locations. This may indicate that LLSVP margins are not stable through time and can be affected by motions within the lower mantle, as suggested in geodynamic models[45] and as predicted in models that explicitly fit the rapid southern motion of the Emperor

seamounts[9]. Alternatively, plumes are not tied to LLSVP margins and, for example, may initiate at the LLSVP margins and migrate towards their interiors over time[46].

In summary, we now have evidence for three well-defined and long-lived hotspot chains across the northern, equatorial, and southern expanses of the Pacific plate, each generated from an independently moving lower mantle plume. Their relative motions to a first order are explained by geodynamic plume motion models and this work provides further evidence that the Hawaiian plume is anomalous in its rapid motion between 80 and 50 Ma. During the same interval, it appears that the Rurutu hotspot maintained a similar distance from the Louisville hotspot, showing that these two mantle plumes have been moving in tandem since 80 Ma. Most importantly, our geodynamic modeling observations indicate that it is highly unlikely that mantle plumes originate from near the base of the upper mantle at 660 km. Instead, our results strongly favor a deeper, unanchored source for the three longest-lived Pacific mantle plumes, potentially near the CMB above the Pacific LLSVP or along its edges.

## Methods

**RR1310 cruise summary.** All samples reported in this study were collected onboard the RV *Roger Revelle* during expedition RR1310. The expedition took place in the summer of 2013 and dredged 43 seamounts in the Tuvalu-Samoa-Tonga region of the Pacific Ocean. This work focuses on the samples recovered from the Tuvalu seamount chain, which previously has been speculated to have been generated from the Rurutu hotspot[13] that currently underlays Arago Seamount in French Polynesia[15]. Along with dredging, seafloor bathymetric mapping was conducted using EM122 multibeam sonar. Maps were generated using the Seamount Catalog program[47]. All samples recovered from this expedition are

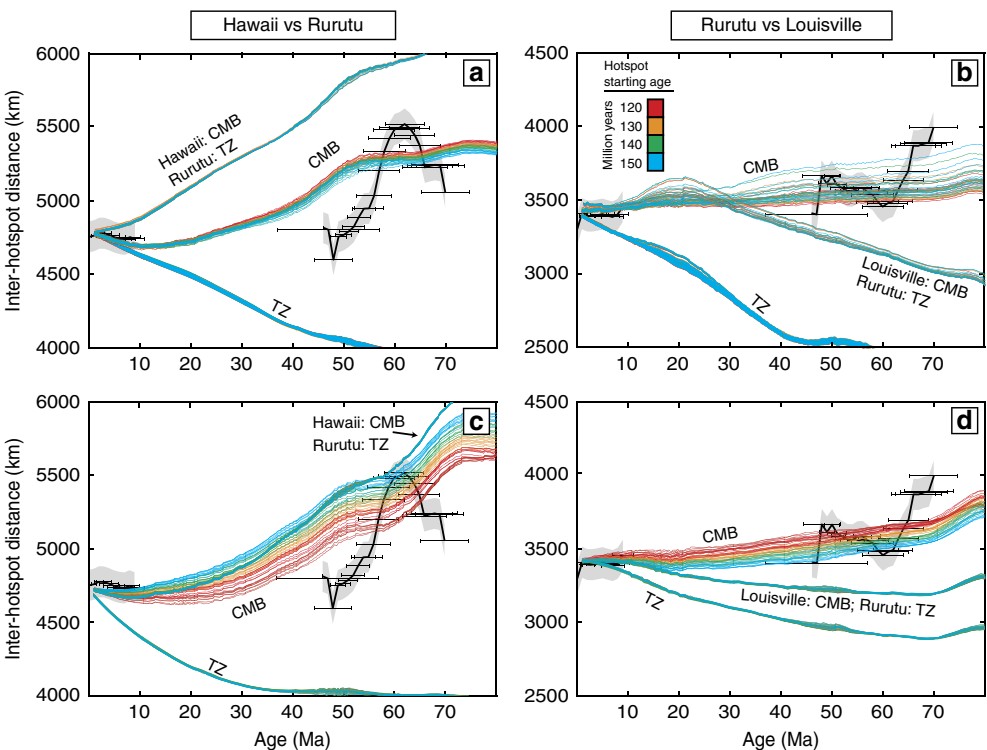

**Fig. 3** The inter-hotspot distances compared against modeled plume distances. The distances are from Fig. 2 with additional modeled plume motion comparison of a Rurutu plume generated from the transition zone (TZ) while Hawaii and Louisville are generated from the core-mantle boundary (CMB). All examples shown here assume mobile plume roots. **a** Hawaii compared against Louisville. **b** Rurutu compared to Louisville. **c** Hawaii compared to Rurutu. **d** Rurutu compared against Louisville. Panels **a**, **b** use the viscosity model of Steinberger and Calderwood[56] and tomography model TOPOS362d1[69] with the same age and buoyancy parameters as Fig. 2. Panels **c**, **d** use the tomography model SMEAN[62] and the viscosity model of Rudolph et al.[61]. In panel **c**, Hawaii has an assumed starting age of 150 Ma and buoyancy of $6 \times 10^3$ kg/s. In panel **d**, the modeled Louisville plume assumes a starting age of 120 Ma and buoyancy of $4 \times 10^3$ kg/s

archived at the Oregon State University (OSU) Marine and Geology Repository (http://osu-mgr.org) and are available upon request.

**Incremental heating $^{40}$Ar/$^{39}$Ar geochronology.** Petrographic analyses were conducted on all potential candidates for age determinations. Rocks that contained fresh phenocrystic phases (e.g. plagioclase, hornblende) were preferentially selected and dated. In absence of useable phenocrysts, the freshest holocrystalline groundmasses were selected for $^{40}$Ar/$^{39}$Ar analyses. When available, multiple phases from a single rock were analyzed. The sample preparation protocol is based on previously established methods for obtaining useable ages from altered submarine lithologies[48,49]. Rocks were first crushed and sieved to grain sizes of either 108–180 μm (more altered) or 212–300 μm (less altered). The grains were then sonicated and rinsed in deionized $H_2O$ prior to drying in a 50 °C oven overnight. The dried grains were magnetically separated in an effort to concentrate individual phases of plagioclase, hornblende, and/or groundmass. Subsequently, the separates were leached in acid for 1 or 2 h each in 3 N HCl, 6 N HCl, 1 N HNO$_3$ and 3 N HNO$_3$ with rinsing in between with ultrapure $H_2O$. Plagioclase separates were additionally sonicated in 5% HF for up to 30 min in order to further remove alteration along the edges of those grains. All samples were then sonicated in ultrapure milliQ $H_2O$ for 1 h prior to drying in an oven at 50 °C. The cleaned separates were then handpicked under a binocular microscope in an effort to create a homogenous separate free of alteration.

Between 3 and 50 mg of each sample was loaded into aluminum packets for irradiation. Flux monitors (FCT-2 sanidine) were loaded at the base of the quartz vial and between every three samples. Secondary age standards for addressing long-term internal reproducibility (AC-2 sanidine) were also loaded with the samples. Samples were irradiated using the Oregon State University TRIGA reactor for 6 h. Incremental heating experiments were conducted for each sample with typically 9 −32 steps for phenocrysts and 35−43 steps for groundmass separates. Blanks were measured at the beginning, end, and at every two to three incremental heating steps of the experiment. For groundmass separates, an increased amount of low temperature steps were undertaken in an effort to drive off alteration-derived atmospheric Ar and reduce recoil effects. Most groundmass plateaus were selected in the intermediate heating range and likely represent the degassing of the interstitial K-rich mesostasis within the groundmass[48].

Samples were loaded into a copper tray, which was brought under an ultra-high vacuum. Separates were precleaned using a low energy beam of a $CO_2$ laser. Extracted gas was exposed to a series of 400 °C, 200 °C and room temperature getters prior to being inlet into an ARGUS VI mass spectrometer. Five masses were analyzed simultaneously with 40, 39, 38, and 37 measured on $10^{12}$ Ω Faraday cups while 36 was measured using an ion-counting CuBe electron multiplier. All ages are normalized (including previously published ages) to a Fish Canyon Tuff (FCT) sanidine age of 28.201 ± 0.046 Ma[50] using the decay constant of 5.530 ± 0.097×10$^{−10}$ yr$^{−1}$ (2σ)[51]. Tuvalu ages were calculated using ArArCALC v2.7.0[52] with errors including uncertainties on the blank corrections, irradiation constants, J-curve, collector calibrations, mass fractionation, and the decay of $^{37}$Ar and $^{39}$Ar. The consistency of the secondary AC-2 sanidine age standard analyzed during the time interval when this research was undertaken is excellent with an age of 1185.2 ± 3.8 ka (2σ; MSWD = 6.62; N = 257/275) and within error with other $^{40}$Ar/$^{39}$Ar laboratories[53].

**Excess argon corrections.** In order for a sample with excess argon ($^{40}$Ar/$^{36}$Ar isochron intercept being statistically higher than the 295.5 atmospheric value) deemed "correctable", we set the conservative conditions that the isochron must produce an MSWD < 2, contain at least 15 consecutive heating steps on the plateau, and contain an intercept uncertainty of <10% (1σ). For all four excess Ar samples reported herein (see Supplementary Information) the excess came at the low temperature heating steps while analyzing a groundmass separate. This is hypothesized to be a function of mantle-derived Ar being retained in the glassy (interstitial) portions of the groundmass upon cooling due to the hydrostatic pressure imposed on the lava flow during cooling at depths[54]. Thus, increased hydrostatic pressure on the Rurutu submarine lava flows may amplify the likelihood of not completely equilibrating with atmospheric argon upon eruption.

**Available hotspot age databases.** This study focuses solely on the three best studied long-lived hotspots in the Pacific: Hawaii, Louisville, and Rurutu (new in this study). All ages used were either $^{40}$Ar/$^{39}$Ar or K/Ar (for young Rurutu seamounts and some Hawaiian seamounts) with preference always given to $^{40}$Ar/$^{39}$Ar ages for a given seamount. Supplementary Fig. 5 displays a satellite bathymetric map of the Pacific Basin with all the seamounts and corresponding ages used in our models shown. All $^{40}$Ar/$^{39}$Ar age determinations were recalculated to same standard and decay constant discussed in the Methods. In an effort to best match the time when the hotspot was located most directly beneath the seamount (e.g. shield building stage), only the oldest age for each seamount was utilized. Supplementary Table 3 shows the corrected ages and references used to generate the best fit hotspot track models discussed below.

**Hotspot track reconstruction model.** Combinations of dated and undated seamounts were used to generate the best fit hotspot location at a given age. Each

geographic track was reconstructed by interpolating along a great circle between individual seamounts on a hotspot track (Fig. 1). The age progression of each hotspot track then was reconstructed using a Monte Carlo approach. Hotspot tracks were divided into segments based on similar slopes of seamount age versus distance from the hotspot. We ran this model 1000 times, randomly removing 20% of the age-constrained seamounts resulting in differing local plate velocities for each run. A piecewise cubic hermite interpolating polynomial function was then used to fit the variable age data to the geographic reconstructed hotspot tracks. The 1000 age progression reconstructions were averaged and a standard deviation was taken. All three hotspot tracks were subjected to the same methods in order to generate the three best-fit age progressive hotspot models. A deficiency of this method is that some areas become more sensitive to removing seamounts, which results in an over estimation of uncertainty at ~47 Ma on the Rurutu track. For the Rurutu track, only seamounts with both HIMU geochemistry and age determinations were used due to the high density of hotspot tracks that comprise the South Pacific Isotopic and Thermal Anomaly (SOPITA) and affected the region[13,32–34].

**Inter-hotspot distances.** The distance between individual hotspots at a given time was calculated using the haversine equation for great circles[55] (Fig. 1; black line). Uncertainties on the distances were calculated using a sum of squares of individual track age uncertainties, including an assumed plume radius of 75 km. In order to further test this method, the distance between a seamount of a given age and the coeval point of a modeled seamount on the compared modeled hotspot track was plotted (Fig. 1; circles). For example, when comparing Rurutu to Hawaii we would calculate the distance between the center of the seamount Laupapa (52.98 Ma) in the Rurutu track and where on the modeled Hawaiian hotspot a contemporary modeled seamount would plot. This method is done for all the seamounts within the three tracks that have age constraints. This provides a rough level of scatter, which is typically less than 200 km and consistent with the uncertainties estimated with the Monte Carlo results. The uncertainty on the rate of relative plume motion is calculated by coupling uncertainty on plume radius (75 km) along with the age uncertainty from the reconstructed hotspot tracks (1σ).

**Comparison against geodynamic plume motion models.** The measured inter-hotspot distances since 72 Ma (Fig. 1) were then compared against a multitude of model runs generated for Rurutu, Hawaii, and Louisville by using a previously published geodynamic mantle convection and plume motion model[20–23]. Using this model, we predicted past geographic locations of the plumes underlying the Pacific lithosphere at any given time (limited in this study to between 80 Ma and the present day). We then used the same great circle equations discussed above to produce modeled inter-hotspot distances.

These Steinberger-style geodynamic models produce variable results depending on a few key parameters briefly summarized here. Present-day mantle densities are inferred from seismic tomography models and used to compute mantle flow assuming Newtonian viscous rheology. Past mantle densities and inferred mantle flow are then computed with backward-advection in the time-dependent flow field; however, in order to account for the increasing uncertainties back in time, backward-advection is limited to the past 68 Ma. For this study the conversion factor for seismic velocity to density anomalies uses model "2" of Steinberger and Calderwood[56]. This scaling factor is reduced to 50% in the upper 220 km of the mantle, such that density anomalies are not incompatible with observations. In addition, an assumed viscosity structure of the mantle is needed to constrain rates of horizontal and vertical motion in mantle flow fields and between the plume and ambient mantle. Starting ages for each plume are estimated as plumes become more advected in mantle flow through time before potentially stabilizing. Plume rising speeds are calculated as in the computations of Steinberger et al.[57] and require an estimate for their buoyancy fluxes as these control the rate and ability for a plume to rise through the mantle. The depth at which a plume is rooted also is vital, because it controls through what mantle flow fields the plume rises and the timescales of ascension. Finally, we should consider whether or not the plume root (e.g. LLSVP) can be moved by the overlying mantle flow or not. A mobile root will increase the plume's ability to move with the motions of mantle flow as opposed to resisting the motions. As upper boundary condition, the models use the absolute plate motions of Torsvik et al.[38], who adopted the rotation poles of Steinberger and Gaina[39] for the Pacific plate in the time period discussed.

In an effort to compare which previously published tomography and viscosity models best fit our new observations, we compared the inter-hotspot distance between the modeled hotspots to the inter-hotspot distances calculated from the observed data (Fig. 1). We choose to focus on the viscosity models A and B of Čížková et al.[58], Mitrovica and Forte[59], the Roy and Peltier[60] three-layer model and VM6 model, models A and B from Steinberger nd Calderwood[56], Steinberger[21] and Rudolph et al.[61]. For the Rudolph, et al.[61] model we used a simplified version of the results with viscosities of 3×10$^{22}$ Pa·s below 1000 km, 4×10$^{21}$ Pa·s for 1000–670 km depth, 4×10$^{20}$ Pa·s for 670–100 km and 1×10$^{22}$ Pa·s above 100 km. In addition, we compared the tomography models of SMEAN[62], GRAND10 (an update of Grand[63]), GYPSUMS[64], S20RTSB[65], S362ANI[66], SAVANI[67], SAW642[68], TOPOS362d1[69], and TX2008[70]. The combination of different plume parameters with various viscosity and seismic tomography models resulted in 1.8 million different outcomes (e.g. the colored lines on Fig. 2). Due to the complications arising from calculating plume motion into deep time (older than 40 Ma) using

modern tomographic models, no inferences on which tomography or viscosity models best fit the observed data are presented. However, plumes rooted above the transition zone consistently did not reproduce first-order observations of relative plume motion directions. In addition, if just the Rurutu hotspot is generated near the transitional zone while the other hotspots are generated near the CMB, then the motions tend not to reproduce first-order observations either (Fig. 3).

**Data availability**. All data for $^{40}Ar/^{39}Ar$ age determinations are available both in the Supplementary Information and in the online EarthRef.org Digital Archive (https://earthref.org/ERDA). All metadata associated with the RR1310 cruise is available upon request from OSU's Marine and Geology Repository (http://osu-mgr.org). Individual model results are available upon request.

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

## Acknowledgements

We are grateful to RV *Roger Revelle* captain Wes Hill and the technical and science crew of the RR1310 "Rurutu" expedition. Dan Miggins, Julie Klath, and Andrea Balbas are thanked for their assistance with $^{40}$Ar/$^{39}$Ar geochronology analyses. This manuscript greatly benefitted from discussions with Susan Schnur, Daniel Heaton, Bob Duncan, and Dave Graham. This project was funded by NSF Grant 1154070 to A.A.P.K.

## Author contributions

K.K. carried out the $^{40}$Ar/$^{39}$Ar age determinations, calculated the relative plume motions and undertook statistical modeling. A.A.P.K., J.G.K. and M.G.J. were PIs on RR1310 expedition. A.A.P.K. supervised all aspects of the research. B.S. provided the geodynamic plume models. V.A.F. provided the Pb isotopic data. All authors contributed to the discussion in this paper.

## Additional information

**Competing interests:** The authors declare no competing interests.

