## [Peer Review File · Nature Communications]

Reviewers' comments:

Reviewer #1 (Remarks to the Author):

The inference gleaned from new Ar-Ar dating synthesized with existing age data showing that a hot spot track geographically between the Hawaiian-Emperor chain and the Louisville chain (combined Gilbert-Tuvalu-Rurutu chain) is very important and provides added insight into the stability and motion between mantle plumes. The observations stand on their own (without extensive interpretation) and can be used as fundamental constraint on geodynamic models. The implication is that indeed the source of the Hawaiian-Emperor chain rapidly migrated to the south and then essentially stops at circa 50 Ma, but not the 'Rurutu' hot-spot chain in the central Pacific did is fundamental. The observations in Figure 1 will provide new targets of mantle flow models and aid in the development new interpretations of LLSVP stability. The observations should be published rapidly in a journal such as this one as different groups are actively working on this problem.

Unfortunately, the modeling presented is problematic and not needed to warrant publication; essentially models cannot explain the rapid slowing down of the Emperors (Fig. 2), despite running $>10^6$ models. Recent state-of-the-art models, such as Ref 12 can fit at the Emperor slow-down, but the discussions of these new models are sidelined. I would recommend, eliminating the models (which don't work anyway) and provide a less biased discussion of models which fit some of the observations much better. The observational sections can essentially be published as is; I do have some minor comments below.

Mike Gurnis

Minor points.

1. Put tick marks (major and minor) on all four sides of the age-distance plots in Fig. 1a. It's then easier for readers to quickly judge the rapid changes in inter-hot-spot motion described in the text.
2. Figure 1, caption says, "White hexagons represent the location of seamounts with Rurutu...". Do you mean blue-filled circles?
3. This sentence is entirely ambiguous and unhelpful "Relative to the tomography model SMEAN43 we find that the viscosity model for Earth's mantle presented in Steinberger and Calderwood 41, Čížková, et al. 44 and Rudolph, et al. 45 best reproduce the observed motions. In contrast, the viscosity models of Steinberger 38, Mitrovica and Forte 46 and Roy and Peltier 4 tend not to reproduce the observed motions." The authors should think more carefully what at the common elements of the viscosity models that fit versus those that do not. Then rewrite "Assuming at tomography model43, flow models assuming viscosity structures that have (state features) 41,44,45 best reproduce hotspot motions versus viscosity models (state features) 38,46,41 do not."
4. I don't believe that I understand the statement, "The third observation is that, when the roots of the plumes are permitted to move with the convective cells within the lower mantle,". This should be rewritten more explicitly.
5. This sentence needs to be rewritten "This indicates: (1) that LLSVP margins may not be completely stable through time and can be affected by the motions within the mantle, as suggested in geodynamic models (53)" by adding, for example, "as predicted in current models that explicitly fit the rapid southern motion of the Emperor seamounts (ref 12)".
6. Regarding, "(2) that plumes may not be tied to LLSVP margins," Do you mean "fixed LLSVP margins"?

Reviewer #2 (Remarks to the Author):

Review of Konrad, Koppers, Steinberger, Finlayson, Konter and Jackson

This manuscript contributes important new data that will improve our understanding of Pacific hotspots. There is a need to address a few points, but overall I strongly recommend publication.

There are three areas in which the manuscript could benefit from improvement: discussion and analyses of relative vs. absolute motions, true polar wander and geodynamic models vs. observations. Also a minor point in this manuscript, but one of great interest, is the question of the Hawaiian-Emperor Bend, addressed in only the supplement. Finally, some uncertainties are needed.

These issues are discussed in some detail below.

Relative vs. Absolute Motions

1. Arguably the most important conclusion—and least based on models--- presented in this manuscript is that simple geometric comparisons based on new ages of hotspot tracks require large relative motion between hotspots. However, the timing of the identified rapid relative motions precisely coincides with the absolute motion (relative to the spin axis) defined by paleomagnetic data. Thus, this new work represents important independent confirmation of the paleomagnetic results, a point that is made, but only deep in to the manuscript (p 5, line 88). This is something that should be mentioned in the abstract because it will be of general interest. To truly understand mantle processes, one needs to move beyond relative motions.

2. The second and related conclusion is that the plumes can have significant differential relative motions and this points to, in some cases, local controls. The potential local controls are nicely summarized in lines 131-132. The authors, however, fail to develop a consideration of absolute motion. I strongly urge the authors to reconsider; I think there is more they can do with their data with respect to interpretations, conclusions and predictions. For example, for the twelve million-year long period between 60 and 48 million years ago, the authors are calling upon 800 and 750 kilometers of relative motion between Hawaii and Louisville and Rurutu hotspots, respectively. Relative to Hawaii, these distances and times correspond to rates of approximately 67 mm/yr and 63 mm/yr for Louisville and Rurutu, respectively. These are very high rates. What do the authors envision is the most likely geodynamic scenario to account for the numbers above? At face value, these rates directly question the analyses of Torsvik et al. (2017) [see more below] who argued for slower absolute rates that are easier to reconcile with geodynamic models. The conclusion is that these geodynamic models are not up to the task.

True Polar Wander

Early in the manuscript the authors introduce the much confused topic of true polar. Specifically, on Line 29 they state "However, the paleomagnetic interpretations may be complicated by true polar wander (refs 10,11)." True polar wander is complicated, but the authors can do more in helping to move the issue forward, as I will explain below.

True polar wander is a rotation of the entire solid Earth relative to the spin axis. If global paleomagnetic data define a coherent rotation, true polar wander is justified.

It should be noted that the Sager and Koppers work (ref 10) was based on seamount paleopoles which are generally considered today to be unreliable (these values are derived from magnetic surveys over seamounts, and therefore viscous remanent magnetizations, multiple polarities, and non-uniform magnetizations come into play). Moreover, the specific true polar wander hypothesis suggested in ref 10, while intriguing, was shown to be inconsistent with other global data (see

Technical Comment by Cottrell and Tarduno, 2000) and therefore an artifact of seamount paleopole data. This was also implied by the unrealistically high rate of motion in the Sager and Koppers work (ref 10), something that has been pointed out in many subsequent studies.

The second reference (#11) reveals another issue with true polar wander: frames of reference. For decades, true polar wander was defined in a fixed hotspot reference frame. When hotspots were shown to be mobile, much true polar wander disappeared. Recently, several papers that again call for true polar wander (but smaller amounts) rely in one way or another on critical reference frame or geodynamic model assumptions. Thus, true polar wander in most recent treatments is a model parameter, and not necessarily a requirement of global paleomagnetic data.

It should be noted that true polar wander for Cretaceous to Paleogene Pacific was tested and found to be inconsistent with observations. Most importantly, the authors' own comparisons of paleomagnetic data from the Hawaiian-Emperor track and Louisville argue against true polar wander. To be specific: Koppers et al. (ref 2) search for rotation poles to satisfy the paleomagnetically-constrained motion of the Louisville and Hawaiian hotspots to test the hypothesis that these motions might represent a whole-scale motion of the Pacific mantle. They excluded the hypothesis, as noted on lines 28-29 of this manuscript. This is also a test of true polar wander because in one hemisphere the motion of that hemisphere's mantle is indistinguishable from a true polar (i.e., entire solid Earth) rotation.

In summary, while "paleomagnetic interpretations may be complicated by true polar wander", studies have tested Pacific true polar wander and found it to be insignificant relative to hotspot motions (e.g. Tarduno and Gee, Nature, 1995; Tarduno, 2007). The paleomagnetic comparison between paleomagnetic data from the Hawaiian and Louisville trend that argues against basin coherent plume motion (Koppers et al. 2012) also independently tests and fails to support true polar wander as a cause of the paleomagnetically-constrained motions, thus supporting prior conclusions in rapid southward Hawaiian plume motion as is also required by the relative motions defined in this manuscript. I suggest that the authors revise their statement on Line 29 to address the internal inconsistencies of a true polar wander hypothesis to explain the observed hotspot traces in the Pacific.

I will also note that a recent study by Torsvik et al. (NatComm, 2017), published while this manuscript was in review, exclude all paleomagnetic data from the Hawaiian-Emperor chain on the basis of the spurious claim that non-dipole fields at the location of Hawaiian will create the apparent fast hotspot motion. There are many inconsistencies in the Torsvik et al. (2017) work; it is disappointing that these were missed by the reviewers. But perhaps it is sufficient to note that the non-dipole field effects are minor relative to the salient observed values, a finding of global analyses of paleomagnetic published for decades. I suggest that the authors take this opportunity to address these new interpretations in light of their data.

Geodynamic Model Assumptions

The conclusions on plume source are well described and justified. Lines 128-131 are excellent in describing the limits in modeling and failure of models to explain sharpness of the Bend. However, I think the authors could still do a little more in explaining their model assumptions, thereby better separating observations and interpretations.

One central issue can be summarized as follows. The entire point of the manuscript is that the geometric relations between the hotspot tracks studies are inconsistent with fixed hotspots and therefore require plume motion. However, for the key new chain, the authors use a fixed hotspot model to help identify seamounts composing the chain (needed because the chain is intermittent). In fact, one might ask, "How can a hotspot reference frame be constructed given the relative

motions described in this manuscript?"

The answer is that in the references cited (e.g. Wessel et al., 2006) grand averages of several Pacific hotspot tracks are taken. Not surprisingly, these fixed hotspot models fit Hawaii poorly, and hence they are internally inconsistent. So to use such models to identify a new hotspot track needs some caveats. The authors should note that these are crude approximations.

It is interestingly, however, that the authors claim that the same result comes from the work of Raymond – presumably based on plate circuits. It would be useful if the authors could show these various predictions so readers could decide for themselves whether the seamounts chosen best define the track (especially important for the oldest part of the trace where seamounts produced by other hotspots may add complications).

Hawaiian-Emperor Bend

There has been much confusion about the Hawaiian-Emperor Bend in large part because many workers have sought to force a plate tectonic solution. The authors note in the supplement (line 236) that a "significant change in plate motion is required at ~49 Ma, which indicates that the Pacific plate did experience a noticeable change in motion concurrent with the Hawaiian-Emperor".

Why does the authors data require a change in plate motion? If they are just tracking relative plume motion, why can't the Rurutu plume have different local controls than Louisville?

The authors go on to list a series of events that don't even correlate with the best dates on the age of the bend (lines 239-242). I suggest they simply remove these lines. If they retain this argument, they will need to include a discussion of the relevant Wright et al. (2016, Geology) study on Pacific plate motion during formation of the Hawaiian Emperor bend and, specifically, Wright's excellent reply to Wilson's comment (2016). The reply is particularly effective in reemphasizing problems with the simple plate motion explanation.

Uncertainties

The authors should quote uncertainties in the critical time and distance values (e.g. distance between Hawaii and Rurutu).

Summary

This is an important paper that contributes new data confirming and extending prior conclusions on the rapid motion of Pacific hotspots.

The following revisions are needed:

- Note in the abstract that the new results support fast absolute plume motion (this is as, or more important, for geodynamics than relative motions)
- Consider doing more with the rates of motion
- Address inconsistency of true polar wander explanations, on basis of these observations
- Note this work does not support the Torsvik et al. (2017) work calling for slower absolute hotspot motion which is model-based

- Revise and clarify use of fixed hotspot models
- Include diagrams of various model predictions for location of Rurutu track
- Revise discussion of Hawaiian-Emperor Bend
- Add uncertainties on age-distance data and related rates

Reviewer Comments:

Reviewers' comments in black.

Our responses are in red.

Reviewer #1 (Remarks to the Author):

The inference gleaned from new Ar-Ar dating synthesized with existing age data showing that a hot spot track geographically between the Hawaiian-Emperor chain and the Louisville chain (combined Gilbert-Tuvalu-Rurutu chain) is very important and provides added insight into the stability and motion between mantle plumes. The observations stand on their own (without extensive interpretation) and can be used as fundamental constraint on geodynamic models. The implication is that indeed the source of the Hawaiian-Emperor chain rapidly migrated to the south and then essentially stops at circa 50 Ma, but not the 'Rurutu' hot-spot chain in the central Pacific did is fundamental. The observations in Figure 1 will provide new targets of mantle flow models and aid in the development new interpretations of LLSVP stability. The observations should be published rapidly in a journal such as this one as different groups are actively working on this problem.

Unfortunately, the modeling presented is problematic and not needed to warrant publication; essentially models cannot explain the rapid slowing down of the Emperors (Fig. 2), despite running $>10^6$ models. Recent state-of-the-art models, such as Ref 12 can fit at the Emperor slow-down, but the discussions of these new models are sidelined. I would recommend, eliminating the models (which don't work anyway) and provide a less biased discussion of models which fit some of the observations much better. The observational sections can essentially be published as is; I do have some minor comments below.

We agree with both Reviewer #1 and #2 that the observational data itself already warrants publication. However, removing the modeling section as Reviewer #1 suggests would entail a major change of focus for the manuscript, which Reviewer #2 didn't ask for. On the contrary, Reviewer #2 is highly positive about this section, our approach and our first order findings, and stated that "The conclusions on plume source are well described and justified. Lines 128-131 are excellent in describing the limits in modeling and failure of models to explain [the] sharpness of the [Hawaii-Emperor] Bend. However, I think the authors could still do a little more in explaining their model assumptions, thereby better separating observations and interpretations".

Reviewer #1 calls the modeling "problematic" as it doesn't fit in particular the sharp inter-hotspot motion around 70-50 Ma, but there are several reasons we find that removal of this part of the paper is not warranted and, on the next page, we explain how we are following the advice from Reviewer #2 in improving this section of the paper:

1.) The overall similarity of predicted hotspot motion obtained with our simplified modeling approach and that of the fully dynamic model of Hassan et al. (2015) can be taken as an indication that our model gets the essential physics right, although it doesn't match the rapid motion and slowdown as well as the Hassan et al. (2015) model. Our model comparisons thus allow us to draw first order conclusions on the behavior of plumes, i.e. are plumes deeply sourced, do the plume roots move or are they anchored, and are the plumes moving in the generally correct directions. Our approach is arguably not adaptive enough to also address more local influences, and in that we agree with Reviewer#1, as the Steinberger-style geodynamic modeling has now been superseded by more modern approaches. However, as we will explain below in point 2.) the Steinberger approach does have some immense advantages over the newer geodynamic modeling that recently have been published.

2.) Our model approach that adapts Steinberger's approach is computationally much less demanding, which makes it possible to run a large number of models, producing up to 1.8 million relative plume motion comparisons, while inserting three plumes, one each for the Hawaii, Louisville and Rurutu hotspots, at the exact same locations in every model run. With a geodynamic model like the one used in Hassan et al. (2015) this would not be possible at all, firstly because each of their runs takes a lot longer, which would require years of computational time to produce the same volume of models analyzed in this study. Secondly, their model runs don't produce plumes in the right places and thus cannot systematically provide reproducible estimates on the contemporaneous plume motions of Hawaii, Louisville and Rurutu, which is the very intent in our paper. In other words, by using the Steinberger geodynamic modeling approach we can now make direct comparisons that are calibrated against the shape of the three longest-lived hotspot chains in the Pacific, which all three are now dated using modern-day $^{40}\text{Ar}/^{39}\text{Ar}$ geochronology. This is novel approach would not have been possible without the availability of the Steinberger code.

3.) Because of the very large number of model runs, we can test a large number of different assumptions and combinations of key parameters, and we can deduct which combination seems to work best. From our modeling we thus can conclude that a mobile plume source in the deep mantle gives the best possible fit, which essentially confirms the Hassan et al. (2015) model. In fact, what is stronger is that we can show that Rurutu hotspot, which has been concluded in previous work to be a shallow minor plumelet, also is a deep plume, robust and long-lived, and thus falls in the same category as the *primary* Hawaii and Louisville hotspot trails. This is an entirely new finding not reported on in other geodynamic modeling papers, such as Hassan et al. (2015), but it strongly underlines the conclusions of the first part of our paper.

Besides, there are also a number of shortcomings of the Hassan et al. (2015) model. As already mentioned, they don't exactly match the present-day hotspot position, and what is presented in their paper is a "best case" singular geodynamic model run. If one would rerun their model using (slightly) different parameters, similar as to our approach, it will be impossible to always generate three plumes (one each for Hawaii, Louisville and Rurutu) in the Hassan models and to

have those appear in reproducible locations, which is critical for us to be able to calculate the correct inter-hotspot distances against which we then can compare our new data in Figure 2. Also, if you look at the cross sections in Figure 2 of Hassan et al. (2015), it appears that their thermochemical pile essentially disappears during their model run as it appears to get entrained into the plumes. Therefore, in the Hassan et al. (2015) model the computed southward motion of the margin, and with it the plume, could be (partially) due to this (unrealistic) model feature, whereas in reality the piles are probably stable for much longer times (e.g. Torsvik et al 2010; Tan et al., 2011; Garnero et al., 2016).

However, to accommodate Reviewer #1 we have reduced the discussion of the modeling results in the manuscript and supplement. We now simply focus on the first-order basic observations and we removed any testing of individual tomography and viscosity models from the paper. We also added to the text (lines 117-125) the following explanation: “Even though the Steinberger geodynamic models³⁵⁻³⁸ are limited by using tomographic models of the modern mantle to predict past large scale mantle flow patterns, the insertion of vertically-buoyant plumes at reproducible locations in every model run allows us to compute tens of thousands of inter-hotspot motion histories. These Steinberger model outcomes are directly comparable to observed inter-hotspot distances between the three longest-lived hotspot chains in the Pacific (Figure 2). The Hassan, et al. ⁹ modeling approach, while being more advanced numerically, lacks this feature and thus cannot be easily calibrated against our new ⁴⁰Ar/³⁹Ar geochronological data from the Rurutu hotspot chain and the observed changes in inter-hotspot distance between Hawaii, Louisville and Rurutu over the last 70 Myrs.”

In addition, we added the following statement to the supplement (lines S354-359) for clarity. “Due to the complications arising from calculating plume motion into deep time using modern tomographic models, no inferences on which tomography or viscosity models best fit the observed data are presented. However, throughout all the results analyzed (1,881,600 individual model comparison results), plumes rooted above the transition zone consistently did not reproduce first order observations of relative plume motion directions.”

Mike Gurnis

Minor points.

1. Put tick marks (major and minor) on all four sides of the age-distance plots in Fig. 1a. It's then easier for readers to quickly judge the rapid changes in inter-hot-spot motion described in the text.

Additional tick marks were added as requested.

2. Figure 1, caption says, “White hexagons represent the location of seamounts with Rurutu...”. Do you mean blue-filled circles?

Changed ‘white’ to ‘blue’.

3. This sentence is entirely ambiguous and unhelpful “Relative to the tomography model SMEAN43 we find that the viscosity model for Earth’s mantle presented in Steinberger and Calderwood 41, Čížková, et al. 44 and Rudolph, et al. 45 best reproduce the observed motions. In contrast, the viscosity models of Steinberger 38, Mitrovica and Forte 46 and Roy and Peltier 4 tend not to reproduce the observed motions.” The authors should think more carefully what at the common elements of the viscosity models that fit versus those that do not. Then rewrite “Assuming at tomography model43, flow models assuming viscosity structures that have (state features) 41,44,45 best reproduce hotspot motions versus viscosity models (state features) 38,46,41 do not.”

The original sentence was removed due to the concerns and ambiguity behind the methods used to determine those conclusions. As both reviewers suggested, the paper is sufficiently strong without these geodynamic modeling interpretations.

Added the following text to the supplement (lines S354-359) for clarity. “Due to the complications arising from calculating plume motion into deep time using modern tomographic models, no inferences on which tomography or viscosity models best fit the observed data are presented. However, throughout all the results analyzed (1,881,600 individual model comparison results), plumes rooted above the transition zone consistently did not reproduce first order observations of relative plume motion directions.”

4. I don’t believe that I understand the statement, “The third observation is that, when the roots of the plumes are permitted to move with the convective cells within the lower mantle,”. This should be rewritten more explicitly.

Original lines 128-130:

New lines 144-146:

Rewrote the sentence to clarify the point as follows: " The third observation is that, when the roots of the plumes are advected with flow in the lower mantle, inter-hotspot motions are generally better represented than when the roots are kept at fixed locations.”

5. This sentence needs to be rewritten “This indicates: (1) that LLSVP margins may not be completely stable through time and can be affected by the motions within the mantle, as suggested in geodynamic models (53)” by adding, for example, “as predicted in current models that explicitly fit the rapid southern motion of the Emperor seamounts (ref 12)”.

See response to statement 6.

6. Regarding, “(2) that plumes may not be tied to LLSVP margins,” Do you mean “fixed LLSVP margins”?

Original lines 137-138:

New lines 144-150:

To address comments 5 and 6 we rewrote the section as follows:

“The third observation is that, when the roots of the plumes are advected with flow in the lower mantle, inter-hotspot motions are generally better represented than when the roots are kept at fixed locations. This may indicate one or both of the following: (1) LLSVP margins are not stable through time and can be affected by motions within the lower mantle, as suggested in geodynamic models⁴⁶ and as predicted in models that explicitly fit the rapid southern motion of the Emperor seamounts⁹; (2) Plumes are not tied to LLSVP margins and, for example, may initiate at the LLSVP margins and migrate towards their interiors over time⁴⁷.

This was done so that the Hassan et al. model is also cited here (Comment 5), and no longer "sidelined" as the reviewer criticizes. However, regarding comment 6, we meant that plumes may not be tied to LLSVP margins, regardless of whether they are fixed or not, so we don't want to add "fixed" here. Besides, we already mention the possibility of the LLSVP margins not being completely fixed under point 1.

Reviewer #2 (Remarks to the Author):

Review of Konrad, Koppers, Steinberger, Finlayson, Konter and Jackson

This manuscript contributes important new data that will improve our understanding of Pacific hotspots. There is a need to address a few points, but overall I strongly recommend publication.

There are three areas in which the manuscript could benefit from improvement: discussion and analyses of relative vs. absolute motions, true polar wander and geodynamic models vs. observations. Also a minor point in this manuscript, but one of great interest, is the question of the Hawaiian-Emperor Bend, addressed in only the supplement. Finally, some uncertainties are needed.

These issues are discussed in some detail below.

Relative vs. Absolute Motions

Arguably the most important conclusion—and least based on models—presented in this manuscript is that simple geometric comparisons based on new ages of hotspot tracks require large relative motion between hotspots. However, the timing of the identified rapid relative motions precisely coincides with the absolute motion (relative to the spin axis) defined by paleomagnetic data. Thus, this new work represents important independent confirmation of the

paleomagnetic results, a point that is made, but only deep in to the manuscript (p 5, line 88). This is something that should be mentioned in the abstract because it will be of general interest. To truly understand mantle processes, one needs to move beyond relative motions.

The following sentence has been added to the abstract: “This result indicates that Hawaii is unique in its strong, rapid southward motion from 60 to 50 million of years ago, consistent with paleomagnetic observations” (lines 10-12).

The second and related conclusion is that the plumes can have significant differential relative motions and this points to, in some cases, local controls. The potential local controls are nicely summarized in lines 131-132. The authors, however, fail to develop a consideration of absolute motion. I strongly urge the authors to reconsider; I think there is more they can do with their data with respect to interpretations, conclusions and predictions. For example, for the twelve million-year long period between 60 and 48 million years ago, the authors are calling upon 800 and 750 kilometers of relative motion between Hawaii and Louisville and Rurutu hotspots, respectively. Relative to Hawaii, these distances and times correspond to rates of approximately 67 mm/yr and 63 mm/yr for Louisville and Rurutu, respectively. These are very high rates. What do the authors envision is the most likely geodynamic scenario to account for the numbers above? At face value, these rates directly question the analyses of Torsvik et al. (2017) [see more below] who argued for slower absolute rates that are easier to reconcile with geodynamic models. The conclusion is that these geodynamic models are not up to the task.

The following absolute plume motion rates are added to the text lines (83-91): “The comparison of the Hawaii to Louisville hotspots (**Figure 1a**) confirms previous observations^{8,31} and shows a large plume divergence (640 ± 106 km) from 60 to 48 Ma, relatively constant inter-hotspot distances between 48 and 15 Ma, and a small decrease in distance starting around 15 Ma. The comparison of Rurutu and Hawaii (**Figure 1b**) shows a similar decrease in hotspot distance (684 ± 106 km) between 60 and 48 Ma, which is consistent with a fast-moving Hawaiian plume. These inter-hotspot distances correspond to a relative rate of motion for the Hawaiian plume of 53 ± 21 km/Myr as compared to Louisville and 57 ± 27 km/Myr as compared to Rurutu during that 12 Myr interval.”

The high plume motion rates are explained as follows: (Lines 91-93) “These rates indicate a faster rate of Hawaiian hotspot motion than previous models have suggested^{34,35}, if Rurutu and Louisville were approximately stationary in this time interval.”

True Polar Wander

Early in the manuscript the authors introduce the much confused topic of true polar. Specifically, on Line 29 they state “However, the paleomagnetic interpretations may be complicated by true polar wander (refs 10,11). “ True polar wander is complicated, but the authors can do more in helping to move the issue forward, as I will explain below.

True polar wander is a rotation of the entire solid Earth relative to the spin axis. If global paleomagnetic data define a coherent rotation, true polar wander is justified.

It should be noted that the Sager and Koppers work (ref 10) was based on seamount paleopoles which are generally considered today to be unreliable (these values are derived from magnetic surveys over seamounts, and therefore viscous remanent magnetizations, multiple polarities, and non-uniform magnetizations come into play). Moreover, the specific true polar wander hypothesis suggested in ref 10, while intriguing, was shown to be inconsistent with other global data (see Technical Comment by Cottrell and Tarduno, 2000) and therefore an artifact of seamount paleopole data. This was also implied by the unrealistically high rate of motion in the Sager and Koppers work (ref 10), something that has been pointed out in many subsequent studies.

The second reference (#11) reveals another issue with true polar wander: frames of reference. For decades, true polar wander was defined in a fixed hotspot reference frame. When hotspots were shown to be mobile, much true polar wander disappeared. Recently, several papers that again call for true polar wander (but smaller amounts) rely in one way or another on critical reference frame or geodynamic model assumptions. Thus, true polar wander in most recent treatments is a model parameter, and not necessarily a requirement of global paleomagnetic data.

It should be noted that true polar wander for Cretaceous to Paleogene Pacific was tested and found to be inconsistent with observations. Most importantly, the authors' own comparisons of paleomagnetic data from the Hawaiian-Emperor track and Louisville argue against true polar wander. To be specific: Koppers et al. (ref 2) search for rotation poles to satisfy the paleomagnetically-constrained motion of the Louisville and Hawaiian hotspots to test the hypothesis that these motions might represent a whole-scale motion of the Pacific mantle. They excluded the hypothesis, as noted on lines 28-29 of this manuscript. This is also a test of true polar wander because in one hemisphere the motion of that hemisphere's mantle is indistinguishable from a true polar (i.e., entire solid Earth) rotation.

In summary, while "paleomagnetic interpretations may be complicated by true polar wander", studies have tested Pacific true polar wander and found it to be insignificant relative to hotspot motions (e.g. Tarduno and Gee, Nature, 1995; Tarduno, 2007). The paleomagnetic comparison between paleomagnetic data from the Hawaiian and Louisville trend that argues against basin coherent plume motion (Koppers et al. 2012) also independently tests and fails to support true polar wander as a cause of the paleomagnetically-constrained motions, thus supporting prior conclusions in rapid southward Hawaiian plume motion as is also required by the relative motions defined in this manuscript. I suggest that the authors revise their statement on Line 29 to address the internal inconsistencies of a true polar wander hypothesis to explain the observed hotspot traces in the Pacific.

The following statement on true polar wander has been added to the text: “These rates indicate a faster rate of Hawaiian hotspot motion than previous models have suggested^{34,35}, if Rurutu and Louisville were approximately stationary in this time interval.” (lines 93-96).

Since true polar wander and its potential (small) contribution to the observations in the paleomagnetic paleolatitude data is not in the scope of this paper, we have left out any detailed discussion in this paper. We believe Reviewer #2 is asking us to add substantially to an interesting but complex discussion, which would make our paper much too long, as is foreshadowed by the lengthy reviewer comments on a very contentious and detailed topic area. However, we fully agree with Reviewer #2 that “Arguably the most important conclusion—and [the] least based on models—presented in this manuscript is that simple geometric comparisons based on new ages of hotspot tracks require large relative motion between hotspots” and this conclusion is directly supporting previous paleomagnetic data, thus showing that, if there is any true polar wander, it is a minor component. Our inter-hotspot distance data are indeed largely model independent, thus providing much harder evidence than conclusions drawn in the complex modeling-intensive approach by Torsvik et al. (2017).

I will also note that a recent study by Torsvik et al. (NatComm, 2017), published while this manuscript was in review, exclude all paleomagnetic data from the Hawaiian-Emperor chain on the basis of the spurious claim that non-dipole fields at the location of Hawaiian will create the apparent fast hotspot motion. There are many inconsistencies in the Torsvik et al. (2017) work; it is disappointing that these were missed by the reviewers. But perhaps it is sufficient to note that the non-dipole field effects are minor relative to the salient observed values, a finding of global analyses of paleomagnetic published for decades. I suggest that the authors take this opportunity to address these new interpretations in light of their data.

As stated above, it is hard to circumvent our “more direct” observations on the inter-hotspot distances, which gives strong evidence of a relatively fast absolute plate motion for Hawaii (both in comparison with Louisville and now also Rurutu) between 70 and 50 Ma. This makes Torsvik’s arguments on non-dipole field effects on paleomagnetic non-consequential as we provide a strong backing for the paleomagnetic paleolatitude data from Tarduno et al (2003).

In addition, we added that our rates of plume motion appear greater than that modeled by Doubrovine et al., (2012) and Torsvik et al., (2017). (Lines 93-96) “These rates indicate a faster rate of Hawaiian hotspot motion than previous models have suggested^{34,35}, if Rurutu and Louisville were approximately stationary in this time interval.”

However, it is important to note that these results are relative in nature and our methodologies cannot produce an absolute rate of Hawaiian plume motion due to the likely nature of plume drift affecting Rurutu and Louisville.

Geodynamic Model Assumptions

The conclusions on plume source are well described and justified. Lines 128-131 are excellent in describing the limits in modeling and failure of models to explain sharpness of the Bend. However, I think the authors could still do a little more in explaining their model assumptions, thereby better separating observations and interpretations.

As described in our response to Reviewer #1 we have improved the explanation on our modeling approach in various places in this paper.

One central issue can be summarized as follows. The entire point of the manuscript is that the geometric relations between the hotspot tracks studies are inconsistent with fixed hotspots and therefore require plume motion. However, for the key new chain, the authors use a fixed hotspot model to help identify seamounts composing the chain (needed because the chain is intermittent). In fact, one might ask, “How can a hotspot reference frame be constructed given the relative motions described in this manuscript?”

The answer is that in the references cited (e.g. Wessel et al., 2006) grand averages of several Pacific hotspot tracks are taken. Not surprisingly, these fixed hotspot models fit Hawaii poorly, and hence they are internally inconsistent. So to use such models to identify a new hotspot track needs some caveats. The authors should note that these are crude approximations.

It is interestingly, however, that the authors claim that the same result comes from the work of Raymond – presumably based on plate circuits. It would be useful if the authors could show these various predictions so readers could decide for themselves whether the seamounts chosen best define the track (especially important for the oldest part of the trace where seamounts produced by other hotspots may add complications).

Regarding geodynamic model assumptions, we agree that constructing a hotspot reference frame given the relative motions described in this manuscript should be a future goal, but would be beyond the scope of this paper.

In the supplement we provided a comparison of the age progression for Rurutu with various different models.

Lines (S199-S213) “To investigate the potential deep mantle sources that fed the Tuvalu volcanoes we first compared the along-track age-distance relationship between the Rurutu-aged HIMU seamounts in the Cook-Australs (ages from: Bonneville et al., 2002; Dalrymple et al., 1975; Diraison, 1991; Duncan and McDougall, 1976; Krummenacher and Noetzelin, 1966; Matsuda et al., 1984; Rose, 2015; Turner and Jarrard, 1982), Tuvalu (this study) and Gilbert Ridge (Koppers et al., 2007; Konter et al., 2008) and a variety of absolute plate motion (APM) models (**Figure S2**). The APM models shown here cover fixed hotspots from Duncan and Clague (1985), the preferred model in Koppers et al. (2001) and WK08-G from Wessel and Kroenke (2008). In addition is the mobile hotspot model PAC-MHS of Steinberger and Gaina (2007) and the global moving hotspot reference model of Doubrovine et al. (2012). Finally,

included is the plate circuit through Antarctica model of Raymond et al. (2000) for the Pacific (Figures S2; S3). With the exception of the plate circuit based APM model of Raymond et al. (2000), both fixed and mobile hotspot models generally path through the Tuvalu region. It is important to note that no one APM model should exactly fit the Rurutu hotspot as the models are based off assumptions on hotspot fixity or modeled Hawaiian/Louisville plume motion. “

In addition we have two additional figures displaying the relative fits of mobile and fixed hotspot tracks. (Pages S11 and S12).

Hawaiian-Emperor Bend

There has been much confusion about the Hawaiian-Emperor Bend in large part because many workers have sought to force a plate tectonic solution. The authors note in the supplement (line 236) that a “significant change in plate motion is required at ~49 Ma, which indicates that the Pacific plate did experience a noticeable change in motion concurrent with the Hawaiian-Emperor”.

Why does the authors data require a change in plate motion? If they are just tracking relative plume motion, why can't the Rurutu plume have different local controls than Louisville?

The authors go on to list a series of events that don't event correlate with the best dates on the age of the bend (lines 239-242). I suggest they simply remove these lines. If they retain this argument, they will need to include a discussion of the relevant Wright et al. (2016, Geology) study on Pacific plate motion during formation of the Hawaiian Emperor bend and, specifically, Wright's excellent reply to Wilson's comment (2016). The reply is particularly effective in reemphasizing problems with the simple plate motion explanation.

As suggested by the reviewer, we removed the discussion on the Hawaiian-Emperor Bend as to how it may relate to other geological events, again because this is outside the scope of this paper.

Uncertainties

The authors should quote uncertainties in the critical time and distance values (e.g. distance between Hawaii and Rurutu).

We have added uncertainties on lines 85-91.

Summary

This is an important paper that contributes new data confirming and extending prior conclusions on the rapid motion of Pacific hotspots.

The following revisions are needed:

- Note in the abstract that the new results support fast absolute plume motion (this is as, or more important, for geodynamics than relative motions)

Added “This result indicates that Hawaii is unique in its strong, rapid southward motion from 60 to 50 million of years ago, consistent with paleomagnetic observations” (lines 10-12).

Added “These inter-hotspot distances correspond to a relative rate of motion for the Hawaiian plume of 53 ± 21 km/Myr as compared to Louisville and 57 ± 27 km/Myr as compared to Rurutu during that 12 Myr interval. On lines 89-91.

- Consider doing more with the rates of motion

Added “These inter-hotspot distances correspond to a relative rate of motion for the Hawaiian plume of 53 ± 21 km/Myr as compared to Louisville and 57 ± 27 km/Myr as compared to Rurutu during that 12 Myr interval. These rates indicate a faster rate of Hawaiian hotspot motion than previous models have suggested^{34,35}, if Rurutu and Louisville were approximately stationary in this time interval.” On lines (89-93)

- Address inconsistency of true polar wander explanations, on basis of these observations

Removed the sentence “However, the paleomagnetic interpretations may be complicated by true polar wander” originally on lines 28-29, because our data is not paleomagnetic in nature we can provide direct evidence on plume motions largely independent of true polar wander.

The concept of true polar wander is a heated topic and not a focal point of this work. Therefore, to keep this paper succinct, we feel it is better to limit the discussion of true polar wander rather than add a significant portion of text to properly explain the current debate.

Added “Our model-independent method, based solely on chain geometry and radiometric ages, therefore shows that significant motion of the Hawaiian plume occurred, supporting the understanding that observed changes in paleomagnetic-derived paleolatitude^{7,8} cannot be solely a result of true polar wander³⁴. Lines 93-96

- Note this work does not support the Torsvik et al. (2017) work calling for slower absolute hotspot motion which is model-based

As discussed above, we added that our rates of plume motion appear greater than that modeled by Doubrovine et al., (2012) and Torsvik et al., (2017). (Lines 91-93) “These rates indicate a faster rate of Hawaiian hotspot motion than previous models have suggested^{34,35}, if Rurutu and Louisville were approximately stationary in this time interval.”

However, it is important to note that these results are relative in nature and our methodologies cannot produce an absolute rate of Hawaiian plume motion due to the likely nature of plume drift affecting Rurutu and Louisville.

We also included the following text (lines 93-96): “Our model-independent method, based solely on chain geometry and radiometric ages, indicates that significant motion of the Hawaiian plume occurred, which supports that observed changes in paleomagnetic-derived paleolatitude^{7,8} cannot be solely a result of true polar wander³⁴.” This text indicates that our results support large plume movement independent of any paleomagnetic arguments or model assumptions.

- Revise and clarify use of fixed hotspot models
See next comment
- Include diagrams of various model predictions for location of Rurutu track

These diagrams are included in the supplements along with Figure S2. In particular we provide an age versus along track distance graph and a map of the Rurutu track with a couple of different absolute plate motion model paths included. Both fixed and mobile APM hotspot models are used and discussed to address both these two comments.

Added lines (S199-S213) “To investigate the potential deep mantle sources that fed the Tuvalu volcanoes we first compared the along-track age-distance relationship between the Rurutu-aged HIMU seamounts in the Cook-Austral (ages from: Bonneville et al., 2002; Dalrymple et al., 1975; Diraison, 1991; Duncan and McDougall, 1976; Krummenacher and Noetzlin, 1966; Matsuda et al., 1984; Rose, 2015; Turner and Jarrard, 1982), Tuvalu (this study) and Gilbert Ridge (Koppers et al., 2007; Konter et al., 2008) and a variety of absolute plate motion (APM) models (**Figure S2**). The APM models shown here cover fixed hotspots from Duncan and Clague (1985), the preferred model in Koppers et al. (2001) and WK08-G from Wessel and Kroenke (2008). In addition is the mobile hotspot model PAC-MHS of Steinberger and Gaina (2007) and the global moving hotspot reference model of Doubrovine et al. (2012). Finally, included is the plate circuit through Antarctica model of Raymond et al. (2000) for the Pacific (**Figures S2; S3**). With the exception of the plate circuit based APM model of Raymond et al. (2000), both fixed and mobile hotspot models generally path through the Tuvalu region. It is important to note that no one APM model should exactly fit the Rurutu hotspot as the models are based off assumptions on hotspot fixity or modeled Hawaiian/Louisville plume motion. “

In addition we have two additional figures displaying the relative fits of mobile and fixed hotspot tracks. (Pages S11 and S12).

- Revise discussion of Hawaiian-Emperor Bend

Removed section 3.4 (original lines 231-246; below) from the supplements as suggested by the reviewer earlier in his comments.

“3.4 Hawaiian-Emperor Style ‘Bend’ in the Rurutu Track

The sharp hotspot track 'bend' at the Hawaii-Emperor intersection has been inferred to represent either a significant change in Pacific plate motion (e.g. Duncan and Clague 1986) or a shift in Hawaiian plume motion (e.g. Tarduno et al., 2003). In order to link the isotopically similar Tuvalu seamounts to Rurutu in the Cook-Austral Islands, a significant change in plate motion is required at ~49 Ma, which indicates the Pacific plate did experience a noticeable change in motion concurrent with the Hawaiian-Emperor. The timing of the Rurutu and Hawaiian-Emperor bends is roughly coincident with the initiation of subduction at the Izu-Bonin-Mariana trench (51 Ma; Reagan et al., 2010), collision of India with Asia (50 Ma; Zhu et al., 2005), and the re-organization of the Australian and Antarctic plates (Whittaker et al., 2007). The coincidence of these events around 50 Ma indicates that plate reorganization was potentially global in scale (e.g. Müller et al., 2016). The Louisville hotspot trail shows minimal evidence for a distinct bend at 50 Ma (e.g. Koppers et al., 2011), which can be explained due to the proposed eastward drift of the plume (e.g. Steinberger and Antretter, 2006; Hassan et al., 2016) subduing any record of a plate motion change (i.e., of the difference of plate motion before and after the change) from ~N-NW (80-50 Ma) to ~W-NW (50-0 Ma). “

- Add uncertainties on age-distance data and related rates

We added the uncertainty (1-sigma) for the distance between hotspots during the crucial 60-48 Ma timeframe. Hawaii to Louisville 640 ± 106 km (Line 85) and Hawaii to Rurutu 684 ± 106 km (lines 87-88). We also added the sentence (lines 89-91) “These distances corresponded to a relative rate of motion for the Hawaiian plume of 53 ± 21 km/Myr as compared to Louisville and 57 ± 27 km/Myr as compared to Rurutu during that 12 Myr interval.”

REVIEWERS' COMMENTS:

Reviewer #1 (Remarks to the Author):

I believe that this paper can now be published as is and I have no further comments.

Reviewer #2 (Remarks to the Author):

The authors have done a very nice job with the revision, tightening the discussion such that it now better focuses on what can be derived from their new data. I reiterate that the simple comparison of the age trends indicating rapid hotspot motion is a fundamental contribution to understanding the dynamics of mantle plumes that will be widely used and cited. It merits publication in Nature Communications.

The point of modeling should be to help us ask new questions and not to merely bolster previously held notions; the latter has sometimes lead to slow progress in understanding Pacific mantle dynamics. It is notable that the authors now more clearly indicate the shortcomings of their modeling but also highlight avenues for future study. They note that the Hassan et al. modeling reproduces features that their own modeling does not (e.g. line 155). With the exception of a need to better explain the differences between modeling runs (see below) I find the modeling section acceptable.

I would ask the authors to make the following minor revisions.

1. Please quote the significance level on the uncertainties on the rates in the main text (are these 1 sigma?). A sentence in the methods or supplementary materials explaining the error budget would improve the manuscript (does the rate uncertainty simply include the 1 sigma errors in ages or does it incorporate an error in hotspot location?)
2. Models: The authors claim they have run 1,881,600 models. The authors should explain to readers what defines a significantly "different" model. That is, one could imagine incremental differences in model parameters that would not be expected to yield new insights. It would be useful if the authors defined some thresholds in their variation of model parameters. This might result in a downward revision in the practical number of truly different models examined. Note, I appreciate that the authors have expressed willingness to supply the models upon request. But if they are to quote such a large number (of models) they should explain it for readers independent of a special request.

Konrad et al. – Manuscript NCOMMS-17-17357A

Reviewer Comments – Second Review:

Reviewers' comments in black.

Our responses are in red.

Reviewer #1 (Remarks to the Author):

I believe that this paper can now be published as is and I have no further comments.

Mike Gurnis

Thank you.

Reviewer #2 (Remarks to the Author):

The authors have done a very nice job with the revision, tightening the discussion such that it now better focuses on what can be derived from their new data. I reiterate that the simple comparison of the age trends indicating rapid hotspot motion is a fundamental contribution to understanding the dynamics of mantle plumes that will be widely used and cited. It merits publication in Nature Communications.

The point of modeling should be to help us ask new questions and not to merely bolster previously held notions; the latter has sometimes lead to slow progress in understanding Pacific mantle dynamics. It is notable that the authors now more clearly indicate the shortcomings of their modeling but also highlight avenues for future study. They note that the Hassan et al. modeling reproduces features that their own modeling does not (e.g. line 155). With the exception of a need to better explain the differences between modeling runs (see below) I find the modeling section acceptable.

I would ask the authors to make the following minor revisions.

1. Please quote the significance level on the uncertainties on the rates in the main text (are these 1 sigma?). A sentence in the methods or supplementary materials explaining the error budget would improve the manuscript (does the rate uncertainty simply include the 1 sigma errors in ages or does it incorporate an error in hotspot location?)

We appreciate the comment and have added the following sentences to the methods section as well as added (1σ) after mentioning the rates on **line 104**.

Lines 297-299: *The uncertainty on the rate of relative plume motion is calculated by coupling uncertainty on plume radius (75 km) along with the age uncertainty from the reconstructed hotspot tracks (1σ).*

2. Models: The authors claim they have run 1,881,600 models. The authors should explain to readers what defines a significantly “different” model. That is, one could imagine incremental differences in model parameters that would not be expected to yield new insights. It would be useful if the authors defined some thresholds in their variation of model parameters. This might result in a downward revision in the practical number of truly different models examined. Note, I appreciate that the authors have expressed willingness to supply the models upon request. But if they are to quote such a large number (of models) they should explain it for readers independent of a special request.

We agree that a clarity on the individual model results would benefit the manuscript. This section was moved from the supplements to the methods section and the lines were adjusted as follows:

Lines 336-338: *The combination of different plume parameters with various viscosity and seismic tomography models, resulted in 1.8 million different outcomes (e.g. the colored lines on Figure 2).*

Additional Editorial Changes

An additional affiliation was added to the author list at the request of B. Steinberger.

The abstract was reduced to <150 words as requested. The abstract now reads as following:

Lines 2-11: *Mantle plumes upwelling beneath moving tectonic plates generate age-progressive chains of volcanos (hotspot chains) used to reconstruct plate motion. However, these hotspots appear to move relative to each other, implying that plumes are not laterally fixed. The lack of age constraints on long-lived, coeval hotspot chains hinders attempts to reconstruct plate motion and quantify relative plume motions. Here we provide $^{40}\text{Ar}/^{39}\text{Ar}$ ages for a newly identified long-lived mantle plume, which formed the Rurutu hotspot chain. By comparing the inter-hotspot distances between three Pacific hotspots, we show that Hawaii is unique in its strong, rapid southward motion from 60 to 50 Myrs ago consistent with paleomagnetic observations. Conversely, the Rurutu and Louisville chains show little motion. Current geodynamic plume motion models can reproduce the first order motions for these plumes, but only when a plume is rooted in the lowermost mantle.*

The manuscript was broken into sections as requested. It is now structured as:

- Abstract
- Introduction
- Results
 - Longevity of the Rututu hotspot chain;
 - Relative inter-hotspot motions
 - Comparisons to geodynamic plume motion models
- Methods
 - RR1310 cruise summary
 - Incremental heating $^{40}\text{Ar}/^{39}\text{Ar}$ geochronology
 - Excess argon corrections
 - Plume motion modeling
 - Sample suite
 - Hotspot track reconstruction model
 - Inter-hotspot distances
- Data availability
- Acknowledgements
- Author contributions
- Conflict of interest.
- References

To smooth the transitions between sections the following text modified from the supplements was moved into the introduction section:

Lines 33-42: *The Rurutu hotspot^{13,14} is currently active beneath the young Arago seamount (~230 ka¹⁵) and contains a distinct isotopic character that varies from HIMU (high $\square = ^{238}\text{U}/^{204}\text{Pb}$ in source component¹⁶) to C (common mantle component¹⁷) that was originally identified as the ‘Atiu Trend’¹⁸ in the Cook-Austral Islands. Here we provide new age constraints on the Tuvalu seamount chain in the mid-west Pacific between 3° S, 175° E and 10° S, 180° (Figure 1). The seamounts and islands in this region are oriented roughly northwest-southeast, located between the Gilbert Ridge and Samoan hotspot chain, and rest upon lithosphere generated during the Cretaceous paleomagnetic super normal (~110 – 85 Ma)¹⁹. The subaerial*

portions of the islands are entirely composed of coral and thus there have been no previous geochemical and geochronological data on basalts reported from these islands.

The following brief paragraph was added at the end of the introduction section to satisfy the following request: “The final paragraph should be a brief summary of the major results and conclusions.” Many of the lines were moved from the supplement.

Lines 43-50: *The new $^{40}\text{Ar}/^{39}\text{Ar}$ age determinations presented here link the Tuvalu seamount chain to the Rurutu hotspot, allowing us to define the motion of the Rurutu plume relative to the previously dated, long-lived Louisville and Hawaiian hotspots. We then compare these relative motions with the outcome from numerical plume motion models²⁰⁻²³ in order to test which geodynamic mantle parameters best reproduce the observed motions. Results indicate that the Hawaiian plume is unique in its rapid motion ~60-50 Myrs ago, while the Rurutu and Louisville plumes appear to be either relatively stationary or moving in tandem. We find that modeled motions can reproduce first-order trends only if plumes are rooted, but not anchored, near the core-mantle boundary.*

The following sentence was added to the beginning of the Results section:

Lines 53-55: *Seamounts from the Tuvalu region, mid-Pacific (Figure 1) were dredged during the RR1310 expedition. In most dredges, basaltic material was recovered that has been variably altered from which a selective number of samples were deemed useable for $^{40}\text{Ar}/^{39}\text{Ar}$ age determinations.*

The word “Supplemental” was changed to “Supplementary” throughout the manuscript and supplementary document.

A conclusory sentence was added to the end of the Longevity of the Rurutu Hotspot Chain section:

Lines 71-72: *These new age constraints confirm that the Rurutu plume is long-lived¹³ and intermittently expressed where lithospheric weaknesses allow melt penetration.*

Figures 1 and 2, along with supplemental figure 6 was edited to use lower case letters to denote panels as requested.

Figure 1 title was changed to: *Inter-hotspot distances as a function of age and the geographic locations of the Hawaiian, Louisville, and Rurutu hotspots.*

Figure 2 title was changed to: *The observed relative inter-hotspot distances through time compared against modeled simulations.*

All numbered lists were removed from the manuscript.

The methods section was heavily revised and most of the sections from the supplement were added to the methods as requested.

Following sections were added to the Methods:

Lines 196-206:

RR1310 Cruise Summary

All samples reported in this study were collected onboard the R/V Roger Revelle during expedition RR1310. The expedition took place in the summer of 2013 and dredged 43 seamounts in the Tuvalu-Samoa-Tonga region of the Pacific Ocean. This work focuses on the samples recovered from the Tuvalu seamount chain, which previously has been speculated to have been generated from the Rurutu hotspot¹³ that currently underlays Arago seamount in French Polynesia¹⁵. Along with dredging, seafloor bathymetric mapping was conducted using EM122 multibeam sonar. Maps were generated using the Seamount Catalog program (<https://earthref.org/SC/>). All samples recovered from this expedition are archived at the Oregon State University (OSU) Marine and Geology Repository (<http://osu-mgr.org>) and are available upon request.

Lines 247-364:

Excess Argon Corrections

In order for a sample with excess argon ($^{40}\text{Ar}/^{36}\text{Ar}$ isochron intercept being statistically higher than the 295.5 atmospheric value) deemed 'correctable' we set the conservative conditions that the isochron must produce an $\text{MSWD} < 2$, contain at least 15 consecutive heating steps on the plateau, and contain an intercept uncertainty of $< 10\%$ (1σ). For all four excess Ar samples reported herein (see Supplementary Information) the excess came at the low temperature heating steps while analyzing a groundmass separate. This is hypothesized to be a function of mantle derived Ar being retained in the glassy (interstitial) portions of the groundmass upon cooling due to the hydrostatic pressure imposed on the lava flow during cooling at depths⁵⁷. Thus, increased hydrostatic pressure on the Rurutu submarine lava flows may amplify the likelihood of not completely equilibrating with atmospheric argon upon eruption.

Plume Motion Modeling

Available Hotspot Age Databases

This study focuses solely on the three best studied long-lived hotspots in the Pacific: Hawaii, Louisville, and Rurutu (new in this study). All ages used were either $^{40}\text{Ar}/^{39}\text{Ar}$ or K/Ar (for young Rurutu seamounts and some Hawaiian seamounts) with preference always given to $^{40}\text{Ar}/^{39}\text{Ar}$ ages for a given seamount. Supplementary Figure 5 displays a satellite bathymetric map of the Pacific Basin with all the seamounts and corresponding ages used in our models shown. All $^{40}\text{Ar}/^{39}\text{Ar}$ age determinations were recalculated to same standard and decay constant discussed in the methods. In an effort to best match the time when the hotspot was located most directly beneath the seamount (e.g. shield building stage), only the oldest age for each seamount was utilized. Supplementary Table 3 shows the corrected ages and references used to generate the best fit hotspot track models discussed below.

Hotspot Track Reconstruction Model

Combinations of dated and undated seamounts were used to generate the best fit hotspot location at a given age. Each geographic track was reconstructed by interpolating along a great circle between individual seamounts on a hotspot track (Figure 1). The age progression

of each hotspot track then was reconstructed using a Monte Carlo approach. Hotspot tracks were divided into segments based on similar slopes of seamount age versus distance from the hotspot. We ran this model 1,000 times, randomly removing 20% of the age constrained seamounts resulting in differing local plate velocities for each run. A piecewise cubic hermite interpolating polynomial function was then used to fit the variable age data to the geographic reconstructed hotspot tracks. The 1,000 age progression reconstructions were averaged and a standard deviation was taken. All three hotspot tracks were subjected to the same methods in order to generate the three best-fit age progressive hotspot models. A deficiency of this method is that some areas become more sensitive to removing seamounts, which results in an over estimation of uncertainty at ~47 Ma on the Rurutu track. For the Rurutu track, only seamounts with both HIMU geochemistry and age determinations were used due to the high density of hotspot tracks that comprise the South Pacific Isotopic and Thermal Anomaly (SOPITA) and affected the region 13,32-34.

Inter-Hotspot Distances

The distance between individual hotspots at a given time was calculated using the haversine equation for great circles⁵⁸ (Figure 1; black line). Uncertainties on the distances were calculated using a sum of squares of individual track age uncertainties, including an assumed plume radius of 75 km. In order to further test this method, the distance between a seamount of a given age and the coeval point of a modeled seamount on the compared modeled hotspot track was plotted (Figure 1; circles). For example, when comparing Rurutu to Hawaii we would calculate the distance between the center of the seamount Laupapa (52.98 Ma) in the Rurutu track and where on the modeled Hawaiian hotspot a contemporary modeled seamount would plot. This method is done for all the seamounts within the three tracks that have age constraints. This provides a rough level of scatter, which is typically less than 200 km and consistent with the uncertainties estimated with the Monte Carlo results. The uncertainty on the rate of relative plume motion is calculated by coupling uncertainty on plume radius (75 km) along with the age uncertainty from the reconstructed hotspot tracks (1σ).

Comparison Against Geodynamic Plume Motion Models

The measured inter-hotspot distances since 72 Ma (Figure 1) were then compared against a multitude of model runs generated for Rurutu, Hawaii and Louisville by using a previously published geodynamic mantle convection and plume motion model²⁰⁻²³. Using this model, we predicted past geographic locations of the plumes underlying the Pacific lithosphere at any given time (limited in this study to between 80 Ma and the present day). We then used the same great circle equations discussed above to produce modeled inter-hotspot distances.

These Steinberger-style geodynamic models produce variable results depending on a few key parameters briefly summarized here. Present-day mantle densities are inferred from seismic tomography models and used to compute mantle flow assuming Newtonian viscous rheology. Past mantle densities and inferred mantle flow are then computed with backward-advection in the time-dependent flow field, however, in order to account for the increasing uncertainties back in time, backward-advection is limited to the past 68 Ma. For this study the conversion factor for seismic velocity to density anomalies uses model '2' of Steinberger and Calderwood⁴⁹. This scaling factor is reduced to 50% in the upper 220 km of the mantle, such that density anomalies are not incompatible with observations. In addition, an assumed viscosity structure of the mantle is needed to constrain rates of horizontal and vertical motion in mantle flow fields and between the plume and ambient mantle. Starting ages for each plume are estimated as plumes become more advected in mantle flow through time before potentially stabilizing. Plume rising speeds are calculated as in the computations of Steinberger, et al.⁵⁹ and require an estimate for their buoyancy fluxes as these control the rate and ability for a plume to rise through the mantle. The depth at which a plume is rooted also is vital, because it controls through what mantle flow fields the plume rises and the timescales of ascension. Finally, we should consider whether or not the plume root (e.g. LLSVP) can be moved by the overlying mantle flow or not. A mobile root will increase the plume's ability to move with the motions of mantle flow as opposed to resisting the motions. As upper boundary condition, the models use the absolute plate motions of Torsvik, et al.⁴⁰, who adopted the rotation poles of Steinberger and Gaina⁴¹ for the Pacific plate in the time period discussed.

In an effort to compare which previously published tomography and viscosity models best fit our new observations, we compared the inter-hotspot distance between the modeled hotspots to the inter-hotspot distances calculated from the observed data (Figure 1). We choose to focus

on the viscosity models A and B of Čížková, et al. 60, Mitrovica and Forte 61, the Roy and Peltier 62 3-layer model and VM6 model, models A and B from Steinberger and Calderwood 49, Steinberger 21 and Rudolph, et al. 63. For the Rudolph, et al. 63 model we used a simplified version of the results with viscosities of 3×10^{22} Pa·s below 1,000 km, 4×10^{21} Pa·s for 1,000-670 km depth, 4×10^{20} Pa·s for 670-100 km and 1×10^{22} Pa·s above 100 km. In addition, we compared the tomography models of SMEAN64, GRAND10 (an update of Grand 65), GYPSUMS66, S2ORTSB67, S362ANI68, SAVANI69, SAW64270, TOPOS362d150, and TX200871. The combination of different plume parameters with various viscosity and seismic tomography models, resulted in 1.8 million different outcomes (e.g. the colored lines on Figure 2). Due to the complications arising from calculating plume motion into deep time (older than 40 Ma) using modern tomographic models, no inferences on which tomography or viscosity models best fit the observed data are presented. However, plumes rooted above the transition zone consistently did not reproduce first order observations of relative plume motion directions. In addition, if just the Rurutu hotspot is generated near the transitional zone while the other hotspots are generated near the CMB, then the motions tend not to reproduce first order observations either (Supplementary Figure 6).

Data Availability

All data for $^{40}\text{Ar}/^{39}\text{Ar}$ age determinations are available both in the Supplementary Data File 1 and in the online EarthRef.org Digital Archive (<https://earthref.org/ERDA>). All metadata associated with the RR1310 cruise is available upon request from OSU's Marine and Geology Repository (<http://osu-mgr.org>). Individual model results are available upon request.

Acknowledgements

We are grateful to R/V Roger Revelle captain Wes Hill and the technical and science crew of the RR1310 "Rurutu" expedition. Dan Miggins, Julie Klath and Andrea Balbas are thanked for their assistance with $^{40}\text{Ar}/^{39}\text{Ar}$ geochronology analyses. This manuscript greatly benefitted from discussions with Susan Schnur, Daniel Heaton, Bob Duncan and Dave Graham. Mike Gurnis and an anonymous reviewer are thanked for comments that greatly improved this manuscript. This project was funded by NSF Grant 1154070 to A. Koppers.

Author Contributions

K. Konrad carried out the $^{40}\text{Ar}/^{39}\text{Ar}$ age determinations, calculated the relative plume motions and undertook statistical modeling. A. Koppers, J. Konter and M. Jackson were PIs on RR1310 expedition. A. Koppers supervised all aspects of the research. B. Steinberger provided the geodynamic plume models. V. Finalyson provided the Pb isotopic data. All authors contributed to the discussion in this paper.

Conflict of Interest

The authors declare no competing financial interests.

The reference section was cleaned up to fit nature format.

The supplementary document was reduced considerably in size. Most of the sections were moved to the methods. The supplement now includes the extended results section for $^{40}\text{Ar}/^{39}\text{Ar}$ analyses and an extra discussion on sources of the Tuvalu seamount. The document still contains six supplementary figures. All references in the supplementary document are now in proper nature format.